# Viroids: Non-Coding Circular RNAs Able to Autonomously Replicate and Infect Higher Plants

**DOI:** 10.3390/biology12020172

**Published:** 2023-01-21

**Authors:** Beltrán Ortolá, José-Antonio Daròs

**Affiliations:** Instituto de Biología Molecular y Celular de Plantas (Consejo Superior de Investigaciones Científicas-Universitat Politècnica de València), 46022 Valencia, Spain

**Keywords:** circular RNA, non-coding RNA, infectious agent, host plant, rolling-circle replication, hammerhead ribozyme, RNA silencing

## Abstract

**Simple Summary:**

Viroids are the smallest infectious agents currently known. Despite consisting of a relatively small RNA molecule that does not code for any protein, viroids manage to reproduce their genomes and completely invade a host plant when they successfully enter into an initial single cell, frequently inducing a disease. This article recalls viroid discovery about 50 years ago and reviews our current knowledge about most aspects of viroid biology, including the structure of viroid molecules and taxonomic classification, the mechanisms of viroid genome replication and movement, how viroids transmit from plan to plant and how they induce disease in the host plants. Finally, the article also reviews recent efforts to transform these infectious agents into tools useful in biotechnology.

**Abstract:**

Viroids are a unique type of infectious agent, exclusively composed of a relatively small (246–430 nt), highly base-paired, circular, non-coding RNA. Despite the small size and non-coding nature, the more-than-thirty currently known viroid species infectious of higher plants are able to autonomously replicate and move systemically through the host, thereby inducing disease in some plants. After recalling viroid discovery back in the late 60s and early 70s of last century and discussing current hypotheses about their evolutionary origin, this article reviews our current knowledge about these peculiar infectious agents. We describe the highly base-paired viroid molecules that fold in rod-like or branched structures and viroid taxonomic classification in two families, *Pospiviroidae* and *Avsunviroidae*, likely gathering nuclear and chloroplastic viroids, respectively. We review current knowledge about viroid replication through RNA-to-RNA rolling-circle mechanisms in which host factors, notably RNA transporters, RNA polymerases, RNases, and RNA ligases, are involved. Systemic movement through the infected plant, plant-to-plant transmission and host range are also discussed. Finally, we focus on the mechanisms of viroid pathogenesis, in which RNA silencing has acquired remarkable importance, and also for the initiation of potential biotechnological applications of viroid molecules.

## 1. Overview of Viroids

Viroids constitute a group of intracellular parasites of higher plants, composed of a small RNA (246 to 430 nt), covalently closed and single-stranded but highly structured, given its high self-complementarity. Their short sequences do not code for any protein in either the viroid RNAs or the complementary strands. They lack a protective protein coat and depend on sequences and motifs in their RNAs to parasitize host plant cell structures in order to replicate autonomously and complete their infectious cycles. Several crop and ornamental plants are among viroids host species; since the infective process leads to host diseases in many cases, viroids are economically relevant as well as interesting from a biological point of view.

## 2. Discovery and Possible Origin of Viroids

The process that led to the discovery of viroids began in the late 1960s with studies that focused on elucidating the causative agent of potato spindle tuber disease [1], which was suspected of viral etiology [2]. Diener and Raymer, and Singh and Bagnall isolated from diseased plants a low molecular weight free RNA, with characteristics of double-stranded RNA (dsRNA), capable of inducing the infection [3,4]. Later it was proposed that the causal agent of this disease must depend entirely on the host enzymes for its replication, as it was composed of a genome smaller than those of known viruses, too small to encode the genetic information necessary for its replication, but it did not need an auxiliary virus to do so [5]. Therefore, the existence of a new type of pathogens, similar but different from viruses and RNA satellites, was proposed. T.O. Diener coined the term “viroid” (virus-like) to describe these RNAs. Although this proposal was not initially well accepted by many scientists, similar causal agents were soon described for citrus exocortis and chrysanthemum stunt, which contributed to its consolidation [6,7]. Later works refined the molecular and mechanistic knowledge of these pathogens.

Viroid origin is an enigma; several hypotheses have been considered. One type of hypothesis suggests that viroids originate from or are the origin of primitive RNA viruses, as well as deriving from introns, transposable elements or other cell RNAs [8,9,10,11]. The recent discovery of the amazing deltavirus diversity [12] and previously unnoticed properties of ambiviruses [13] may support the virus-viroid evolutionary relationship. A completely different hypothesis is that viroids and other current catalytic RNAs are remnants of the “RNA world” [14,15]. In this hypothesis, RNA was the basis of life, given its ability to store information and catalyze its own synthesis before the unfolding of these functions in DNA and proteins. Existing viroids can no longer replicate on their own, possibly having lost that function when they became strict plant parasites. Currently, viroids are classified, together with satellite RNAs, defective interfering particles, and prions, as subviral agents and are considered the smallest infectious agents described to date. It has been proposed that viroids and viroid-like satellite RNAs (some of them previously known as virusoids) have a monophyletic origin, with the family *Avsunviroidae* acting as an evolutionary link between them [14,16,17], although this proposal is controversial.

## 3. General Structure and Phylogenetic Classification of Viroids

Currently, 33 different viroid species and several sequence variants have been biologically and molecularly characterized. Based on structural characteristics and their impact on biological properties, viroids have been classified by the International Committee on Virus Taxonomy (ICTV) into two families (Table 1). The family *Pospiviroidae*, named after its type species, potato spindle tuber viroid (PSTVd) [18], to which most viroids belong, adopt rod-shaped structures containing conserved sequences and structural motifs: the central conserved region (CCR) and the terminal conserved region (TCR) or the terminal conserved hairpin (TCH; Figure 1A). The sequence of the CCR and the presence or absence of TCR and TCH allow the members of this family to be classified into five genera (Table 1). Five distinct domains have been mapped in these viroids [19]: the central domain (C), containing the CCR and flanked by the pathogenic (P) and variable (V) domains to its left and right, respectively, and two terminal domains, right (TR) and left (TL), the latter containing the TCR or TCH (Figure 1A). Although they are named by specific functions, there is a more complex correlation between different parts of the viroid genome and the biological functions they perform [20,21,22]. These viroids replicate and accumulate in the nucleus [23,24,25,26,27] by an asymmetric rolling circle mechanism.

The family *Avsunviroidae* on the other side, is much smaller. Named after its type species, avocado sunblotch viroid (ASBVd) [28], they do not contain CCR or other conserved motifs typical of the former family, but they contain functional hammerhead ribozymes in the RNA of both polarities (Figure 1B). These viroids replicate and accumulate in chloroplasts [26,29,30,31] by a symmetric rolling circle mechanism. Three of its members, peach latent mosaic viroid (PLMVd), chrysanthemum chlorotic mosaic viroid (CChMVd), and apple hammerhead viroid (AHVd) [32,33,34], have a branched conformation stabilized by kissing-loops and pseudoknots, and are classified in a single genus (*Pelamoviroid*), while ASBVd and the eggplant latent viroid (ELVd) adopt quasi-rod-like conformations [28,35]. These form genera with a single member currently described: *Avsunviroid*, characterized by a high content (62%) in A+U, distinctive among the other viroids [36], and *Elaviroid*, with intermediate properties between the previous genera [35] (Table 1 and Figure 1B).

## 4. Viroid Replication

A rolling circle mechanism was proposed for viroid replication, with differences between both families [37,38]. This proposal was based on (i) the non-detection of homologous DNAs in infected tissues [39,40,41], (ii) the circular nature of viroids [42,43] and (iii) the presence of longer-than-unit RNAs, apparently with tandem repeats of both polarities (by convention, the + polarity is assigned to the most abundant circular RNA), in infected plants [37,38,44,45,46,47].

Plants infected with PSTVd do not accumulate circular forms of − polarity [48,49], so the replication is restricted in this viroid and apparently the other members of its family to the asymmetric variant of the rolling circle mechanism (Figure 2A). Circular RNAs (+) are repeatedly transcribed, producing linear concatemers of RNAs of complementary (−) polarity. These RNAs enter directly into a new replication cycle, generating oligomers of + polarity, which are processed (cleaved and ligated) by host enzymes to generate circular (+) monomers of the viroid. On the other hand, *Avsunviroidae* members follow the symmetric pathway (Figure 2B). Circular monomers (+) are repeatedly transcribed, producing viroid concatemers of − polarity. The self-cleavage activity of the hammerhead ribozymes in the concatemer generates monomeric units that are circularized by host factors, resulting in circular intermediates of − polarity which can enter a new round of replication to generate more circular (+) viroids.

The first step in the replication of the *Pospiviroidae* family members is their entry into the nucleus, which appears to be dependent on interaction with host factors [50]. The participation of a bromodomain-containing protein 1 (Virp1) in the process has recently been demonstrated in citrus exocortis viroid (CEVd) [51]. Virp1 can also bind efficiently to PSTVd [52,53,54,55], interacting with a C-loop conserved in nuclear-replicating viroids [56]. This protein of unknown function contains a nuclear localization signal, and it localizes in such organelle [53,55], being also able to mediate the nuclear import of the satellite RNA of cucumber mosaic virus [57], also containing a C-loop [56]. These latter authors also showed the likely involvement of importin alpha-4 (IMPa-4) in the viroid trafficking process. However, CEVd can also be imported independently of Virp1, while additional nuclear localization domains have been described in PSTVd, in the upper strand of the CCR and/or hairpin I [58]. Thus, additional cell factors and viroid signals may mediate the import process. For instance, hop stunt viroid (HSVd) contains open reading frames encoding short peptides with nuclear localization signals and is associated with polysomes, raising a possible novel viroid trafficking strategy [59].

Replication takes place specifically in the nucleoplasm [27]. Viroids of the family *Pospiviroidae* hijack the host DNA-dependent RNA polymerase II (Pol II) [60,61,62], an ability apparently shared with the complementary (-) strand [63]. The Pol II involved in this process, however, has a remodeled architecture with a reduced number of components in contrast to the polymerase complex on DNA templates [64]. In the case of PSTVd, the transcription factor TFIIIA-7ZF is also required [65], being essential for the polymerase to use RNA as a template, while other canonical factors of general transcription do not participate, suggesting different transcription machinery [64]. The binding sites of Pol II and TFIIIA-7ZF are found in the left terminal region of the PSTVd (+), where the transcription start site is consequently found (nucleotide U359 or C1) [66]. A study [67] suggests that the polymerase recognizes the general rod structure between loops 1–5 rather than its specific sequence, while TFIIIA-7ZF has been mapped to the lower strand between nucleotides 331–347 (loops 3–5) [65]. The CCR is also essential for PSTVd replication. Loop 15 can adopt a Loop E structure characterized by 5–6 consecutive non-Watson–Crick base pairs. This structure is also present in the 5S rRNA, mediating its binding to cellular proteins, such as TFIIIA or the ribosomal protein L5 (RPL5). PSTVd loop E interacts with RPL5 [68], affecting its ability to regulate TFIIIA splicing and favoring the production of the 7ZF variant over the 9ZF, thus benefiting its own transcription [69]. It remains to be resolved whether the processes described with PSTVd are general to other members of its family.

The multimeric strands of both polarities produced during replication have different sublocations. Those of − polarity remain anchored in the nucleoplasm, giving rise to more + multimers; those of + polarity are selectively transported to the nucleolus [27], an organelle in which several cell RNAs are processed. Therefore, the existence of some transport mechanism capable of discerning between both polarities is expected. In this sense, it has been proposed that Loop E, which only occurs in PSTVd (+), is involved in this transport through its interaction with the RPL5 protein, which is related to the movement of ribosomal RNAs [68].

The + oligomers are then cleaved into monomeric units. The cleavage takes place between the nucleotides G96 and G97 in the upper strand of the CCR in PSTVd, and in equivalent sites in other viroid species, always between two G. The sequence of the upper strand of the CCR, together with a short flanking inverted repeat, forms a stem-loop structure with a central CG-rich region and a terminal YCGR tetraloop (hairpin I). Two consecutive hairpins in + oligomers interact via kissing loops to form a quasi-double-stranded structure that is recognized and processed by a type-III RNase, which cleaves at the hairpin loops (now a dsRNA region) of the two units at once, releasing a monomeric linear unit of the concatemer. Although the enzyme responsible for this cleavage is formally unknown, current evidence points to the involvement of a host RNase III since these act on dsRNA, and the viroid cleavage generates RNA termini expected for these enzymes: 2-nt overhangs 3′ ends with 5′-phosphomonoester and 3′-hydroxyl terminal groups [70,71]. Upon cleavage, the monomers likely rearrange into rod-like structures, stabilizing the new 3′ and 5′ ends by base pairing with the lower strand, while loop 15 acquires the abovementioned Loop E structure [70]. The host DNA ligase 1, whose usual substrate is DNA and consumes ATP, recognizes and ligates the 5′-phosphomonoester and 3′-hydroxyl ends of the linear replicative intermediate, both in vitro with a recombinant enzyme produced in *Escherichia coli* and in vivo, as suggested by silencing assays [72]. However, the details that mediate the recognition of the replicative intermediate by the enzyme are currently unknown.

On the other hand, the viroids of the family *Avsunviroidae* are the only infectious agents able to enter the chloroplast, where they replicate in the thylakoid membrane [26,73], although the specific trafficking mechanism of these pathogenic RNAs to the chloroplast is unknown. Viroid import seems to be mediated by a viroid localization signal, either sequences or specific structural motifs, which in ELVd have been mapped in the left terminal region (nucleotides 52–150) [74]. Nuclear-expressed transcripts containing these sequences are efficiently transported to chloroplasts, leading these authors to hypothesize an initial step of the viroid infection in which the ELVd is transported from the cytoplasm to the nucleus prior to being exported to the chloroplast. It has been shown that a region of ELVd (nt 16–182) can effectively mediate transcript import into the nucleus; interestingly, this region is partially overlapping with that required for its import into the chloroplast [75]. The cellular factors involved in the intracellular movements of the *Avsunviroidae* are, however, unknown.

It has been proposed that the nuclear-encoded chloroplastic RNA polymerase is the main host factor involved in the replication of these viroids [76,77]. Less known are the mechanisms by which the enzyme is recruited since the transcription start sites are not conserved between species. In ASBVd, this site is located at U121 and U119 in the + and − RNA, respectively, in the AU-rich right terminal loop of the predicted quasi-rod-like structure of both polarities [76]. In PLMVd, positions A50/C51 and A284/A286 have been determined as the transcriptional start for the + and − strands, respectively [78,79], both located in short stems within conserved hammerhead ribozymes motifs. Although it is speculated that in both ASBVd and PLMVd, specific promoter sequences are necessary for polymerase recognition, the involvement of structural motifs cannot be ruled out. Such is the case of the ELVd, in which these sites (U138 and A48 for the + and − strands, respectively) are not related at the sequence level. Thus, it has been proposed that the polymerase (and/or accessory factors) hijack is dependent on some common but unknown structural feature [80].

The linear concatemers of both polarities are processed by hammerhead ribozymes present in their sequences, generating viroid linear monomeric units without the need for host enzymes [32,33,34,35,81]. Hammerhead ribozymes are small RNA domains with autocatalytic activity. First discovered in satellite RNAs [82] and shortly thereafter in viroids [81], they are more widely distributed than initially anticipated, having been identified in all domains of life [83,84,85]. Structurally, hammerhead ribozymes are composed of three stems (named Helix I, II and III), which may or may not be capped by terminal loops, all surrounding a set of 15 highly conserved nucleotides that mediate catalysis. The ribozymes of all family *Avsunviroidae* viroids are type III, as this stem houses the 5′ and 3′ ends of the ribozymes. Despite their name, derived from the original two-dimensional representations [81,86], ribozymes fold into a γ-shaped structure in which stems I and II establish with each other essential interactions for efficient catalysis under physiological conditions [87,88,89]. These interactions are stabilized by divalent metal ions, usually Mg^2+^; its additional involvement in catalysis is currently discussed [90,91,92,93,94]. In any case, hammerhead ribozymes induce cleavage in RNA through a transesterification reaction that converts a 5′,3′-phosphodiester bond into a cyclic 2′,3′-phosphodiester, also generating a 5′-hydroxyl end. The process is potentially reversible, although the efficiency of the reverse reaction in viroid ribozymes is highly variable and generally low [95,96]. Furthermore, the ASBVd + RNA interacts with two chloroplast RNA-binding proteins, PARBP33 and PARBP35, usually involved in the stabilization, maturation and editing of chloroplast transcripts [97]. PARBP33 acts as an RNA chaperone for the viroid, facilitating the self-cleavage of viroid oligomers in vitro and possibly in vivo.

After the cleavage, the monomers are efficiently circularized by the chloroplastic isoform of the host tRNA ligase, at least in the ELVd, as has been shown both in vitro and in vivo [98]. The main function of this enzyme is to ligate the 5′-hydroxyl and 2′,3′-cyclic phosphodiester ends generated with the excision of introns in pre-tRNAs during the tRNA maturation process [99]. Its function in viroid processing can be replaced by an enzymatic activity of the unicellular green algae *Chlamydomonas reinhardtii* [100,101], while eggplant tRNA ligase can process the other members of the family in vitro [98]. Thus, suggesting both the involvement of this kind of enzyme in the processing of all the *Avsunviroidae* and a conserved mechanism of enzymatic recruitment and processing among them. The exact nature of the viroid-enzyme interaction is unknown, although the quasi-rod-like structure in the central part of the ELVd (containing the ligation site in an internal loop) appears to be necessary for ligation [98,101]. Other domains outside this region, however, appear to be not necessary for circularization [102,103]. In this sense, the role of the hammerhead ribozyme domain has been proposed as a mediator of ELVd-ligase recognition in addition to its role in the monomerization of replication concatemers [104].

## 5. Movement of Viroids within the Plant

The viroid progeny must leave the organelle where replication occurs to colonize the rest of the plant, developing a systemic infection. Viroid spread occurs proximally between cells symplastically connected by plasmodesmata [105] and through the phloem in long-distance transport [106,107] (Figure 3). However, exceptions to this have been described, such as some citrus viroids in which movement through phloem is restricted [108]. Together with the existence of mutations that specifically affect systemic infection [21,109], this suggests that the movement depends on interactions with host cellular components.

Several host factors have been proposed to be involved in viroid movement, such as the chaperone-type cucumber phloem protein 2 (CsPP2), which is the most abundant component of cucumber phloem exudate. It forms a ribonucleoprotein complex with HSVd in vitro [110] and spread the infection through intergeneric grafts, suggesting its contribution to the long-distance phloem trafficking of HSVd [111]. The same group described two additional phloem proteins which are translocatable through intergeneric grafting (a phloem-specific lectin and an unidentified 14 kDa protein) and able to bind ASBVd, suggesting that similar mechanisms could govern the expansion of chloroplast viroids [112]. On the other hand, silencing a *Nicotiana tabacum* phloem protein of unknown function (Nt-4/1) seems to enhance PSTVd transport to young developing leaves [113,114] evidencing its possible role in the vascular movement of the viroid, although how it does so is unknown. Other authors also proposed the role of small RNAs derived from loops 7 and 8 of PSTVd in movement regulation by silencing CalS11 and CalS12, callose synthases that regulate plasmodesmata function by reducing the transit space through callose deposition [115]. Callose-mediated plasmodesmata size exclusion limit has already been related to viral expansion [116,117,118]. Whether movement through plasmodesmata occurs as free RNAs or is associated with plant proteins remains unsolved.

In addition to these host factors, several PSTVd RNA motifs have been related to its movement, being common for specific motifs to mediate transport from or to specific areas of the plant, possibly by interacting with different factors [21,22,109,119,120,121].

## 6. Host Defense and Pathogenesis

During their infectious cycle, viroids must be able to interact with various host factors while overcoming the plant’s defensive strategies to stop pathogenic infection (Figure 3). The dependence of viroids on cellular factors to complete its biological cycle make it likely that hijacking host resources may be a direct and main cause of the phenotypic effects of the infection. However, additional causes for symptom development may explain the lack of linearity between titer and symptoms (especially considering the existence of latent viroids that, despite being asymptomatic, reach significant concentrations in the infected tissue) as well as the effect of specific nucleotide changes able to transform mild strains into aggressive strains.

The almost dsRNA structure of viroids of both polarities, and potentially its dsRNA replication intermediates, make them ideal for the generation of RNA interference (RNAi) responses. RNAi describes a series of highly conserved mechanisms in eukaryotes that regulate gene expression and protect against exogenous and endogenous genetic elements, such as viruses or transposons. RNAi is triggered by small RNAs, usually dsRNA, with high sequence homology to the RNAs to be silenced at transcriptional or post-transcriptional levels via epigenetic modifications in DNA and histones that repress the transcription process and mRNA degradation or translational repression. Several studies have detected viroid-derived small RNAs (vd-sRNAs) of both polarities in infected plants, first in PSTVd [122,123] and later in multiple viroid species of both families [124,125,126,127]. Viroids seem to be substrates for degradation via host RNAi defense. The viroid titer is reduced, and the onset of infection symptoms is delayed by the overexpression of Argonaut proteins [128] or by the experimental introduction of vd-sRNA [129,130,131,132]. Viroid overaccumulation is achieved by silencing RNA-dependent RNA polymerases (RDR) 1 and 6, responsible for generating small secondary interfering (siRNAs) [133,134,135], as well as in co-infections with viruses that express silencing suppressors or through the ectopic expression of these suppressors [136]. Similarly, salicylic and gentisic acids appear to enhance the resistance against CEVd in tomato plants by inducing factors that mediate RNA silencing [137]. Despite this evidence, the resistance of mature viroids against RNAi-mediated degradation has also been described [133]. Unlike plant viruses, viroids do not express silencing suppressors. This resistance thus must reside in the viroid compact secondary structure, its association with proteins that prevent their recognition by the RNAi machinery and/or the fact that viroids of both families replicate in organelles where RISC is not that active [133,138,139,140], particularly in the chloroplast, in which the RNAi machinery has not been detected [141]. It is assumed that chloroplastic viroids produce vd-sRNAs during transit through the cytoplasm before reaching this organelle [142].

An important part of viroid pathogenicity derives from the generated vd-sRNAs that can be directed against host mRNAs and trigger the induction of disease symptoms. This hypothesis, which was raised on a theoretical framework [140], was initially demonstrated with the cucumber mosaic virus Y RNA satellite [143] and later in a PLMVd variant that induces extreme leaf chlorosis or peach calico [144]. This conspicuous symptom only occurs if the viroid sequence variant contains the insertion of a specific 12–13-nt hairpin [145]. Two vd-sRNAs derived from the peach calico-associated insert are homologous to the mRNA encoding the chloroplastic heat shock protein 90 (cHSP90). Thus, vd-sRNAs may induce mRNA degradation and promote chloroplast destabilization, leading to peach calico symptoms [144]. Similar observations of vd-sRNA’s involvement in the downregulation of host genes have been reported in various viroids [146,147,148,149,150]. Notably, vd-sRNAs derived from the virulence-modulating region of PSTVd induce the silencing of a potato transcription factor (StTCP23), inducing the conspicuous spindle tuber symptom [150]. Secondary, trans-acting, phased vd-sRNAs have also been proposed, thus expanding the repertoire of silencing targets [151]. Interestingly, the distribution of vd-sRNAs is not uniform throughout the viroid RNA, but rather vd-sRNAs are concentrated in specific regions of the RNA molecules of both families, many of which had been previously described as pathogenicity determinants [123,126,127,131,152,153]. It is likely that the secondary structures of these regions are more susceptible to being processed by the RNAi machinery [124,131,153].

PSTVd and CEVd also induce the expression of genes of the RNA-dependent DNA methylation (RdDM) pathway in tomatoes [149,154], and members of both families can induce transcriptional silencing by methylating their own transgene [139,155,156,157,158]. In addition, trans-methylation of the partial sequence of PSTVd has been achieved experimentally after infection with tomato apical stunt viroid (TASVd), with which it shares some sequence homology [157], and the methylation of some promoters of endogenous genes has been described after PSTVd infection [149,159]. However, the molecular basis of host gene methylation and the functional impact for both the plant and the viroid need to be clarified [160]. Direct interactions have been described between HSVd and histone deacetylase 6 (HDA6), reducing its activity and promoting epigenetic alterations [161]. It has been hypothesized that this interaction favors the spurious recognition of the viroid as an RNA template for replication and has been related to the hypomethylation of the 5S rRNA gene and transposable elements, increasing its transcription [162,163,164]. In this sense, transcriptomic studies have shown extensive changes in gene expression as a result of nuclear [149,165,166] and, to a lesser extent, chloroplastic viroid infection [167]. Infection of orange trees with citrus dwarfing viroid (CDVd) even produces differential expression alterations in the scion and rootstock [168]. Other global effects have also been observed with PSTVd infection, such as the deregulation of long non-coding RNAs, alteration of microRNA and phasiRNA function, and changes in the splicing pattern of coding transcripts [165]. In this regard, the PSTVd interacts with at least one splicing factor, RPL5, interfering with its function [68,69]. However, it is unknown whether this interaction can induce the described effects or whether interactions with other regulators are required instead.

On the other hand, affecting the translational machinery seems to be an important mechanism of viroid infection. In addition to the transcriptional reactivation of rRNA genes and the PSTVd-RPL5 interaction, it has been described that members of the family *Pospiviroidae* and/or derived RNAs interfere with the activity of the eukaryotic elongation factor 1 [169], the maturation of the 18S subunit [170], repress the ribosomal protein S3a [147] and induce the ribosomal stress response [171].

Finally, it is speculated that, as occurs with viral dsRNAs [172], the almost dsRNA structure of viroids or their replicative intermediates are recognized as pathogenic molecular patterns by the plant immune system. The induction of several proteins related to this process during viroid infection has been described [149,171], and thus, the immune response could be partly responsible for viroid symptomatology. In this sense, it has been proposed, as for viral genomes, that post-transcriptional modifications on viroid RNA might prevent its detection by host immunity mechanisms [173]. Given the dynamic complexity of the host-viroid interaction during the infection process, a recent study provides an overall vision that gives temporal relativity to many of the abovementioned host changes [174].

## 7. Host Range and Symptoms

Most viroids infect dicotyledonous plants, with some exceptions, such as the coconut cadang-cadang viroid (CCCVd), the coconut tinangaja viroid (CTiVd), or the tentative Dendrobium viroid (DVd) [175], which infect monocots. Some viroids, such as HSVd and PSTVd, have a wide host range, while others, such as *Coleus blumei* viroids and those of the family *Avsunviroidae*, are mainly restricted to their natural hosts [176]. Generally, members of the family *Pospiviroidae* produce late, nonspecific, systemic symptoms. Those attributed to PLMVd and other members of the family *Avsunviroidae* are, on the contrary, earlier, specific, and local [176]. Pathogenicity depends on the genomes of both the viroid and the host plant, as well as the environmental conditions. Viroids cover a wide range of symptoms (Figure 4), from asymptomatic infections to those that induce plant death, and in general, can be considered similar to those induced by viruses. At the macroscopic level, viroids induce epinasty and chlorosis of the leaves, deformation in flowers, fruits and reserve organs, stem and bark cracking, growth retardation, dwarfism, etc. At the subcellular level, they induce malformations of cell walls and chloroplasts, formation of plasmalemmasomes and electron-dense deposits in the cytoplasm and chloroplasts [142].

Recent research has identified viroid-like RNAs, possibly viroids according to the described features, infectious and inducing symptoms in filamentous fungi [177]. This observation is in line with other reports in which viroid or viroid-like RNAs were associated with fungi [178,179,180,181], although some of these reports are controversial [182].

**Figure 4 biology-12-00172-f004:**
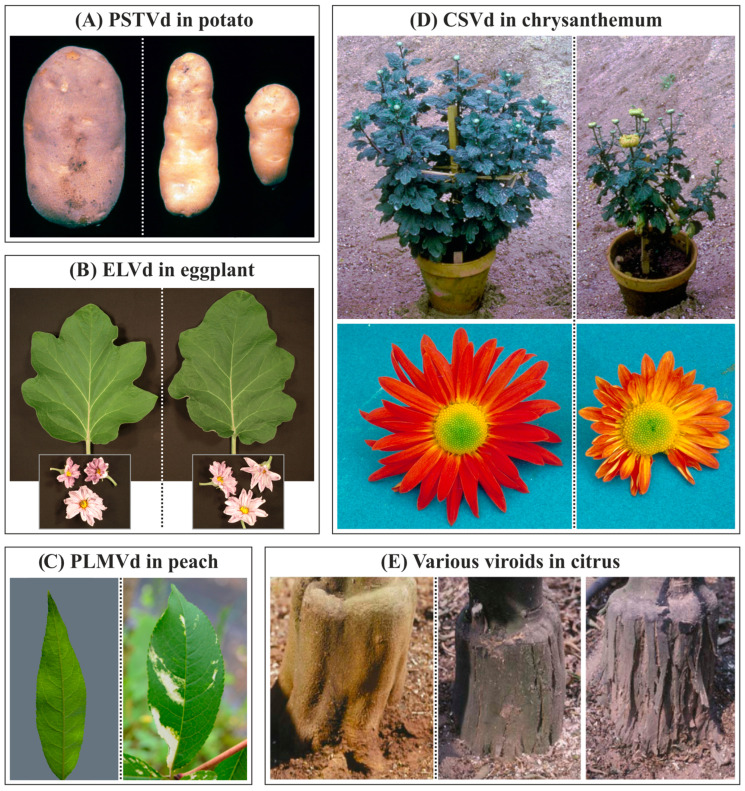
Typical symptoms of viroid infection in various crops. In all cases, mock-inoculated or symptomless plants are on the left, and viroid-infected plants are in the center and/or on the right. (**A**) PSTVd induces potato tuber malformations (image modified from original credited to William M. Brown Jr., Bugwood.org, accessed on 16 January 2023). (**B**) Symptomless infections induced by ELVd in eggplant (cv. Black Beauty). (**C**) Peach calico-inducing variants of PLMVd induce severe chlorosis in peach leaves (left image adapted from https://doi.org/10.3389/fpls.2012.00288, accessed on 16 January 2023; right image modified from original credited to H.J. Larsen, Bugwood.org). (**D**) CSVd infection induces stunting and earlier blooming in chrysanthemum (top), resulting in flower breaking and deformation (bottom) (top image modified from original credited to J. Dunez, Bugwood.org, accessed on 16 January 2023; bottom image modified from original credited to European and Mediterranean Plant Protection Organization, Bugwood.org, accessed on 16 January 2023. (**E**) Co-inoculation of citrus trees with CBLVd and CDVd induces symptomless infections in trees (left), while co-inoculation of CEVd and CBCVd induces bark scaling characteristic of CEVd infection (center) or severe bark cracking characteristic of CBCVd infection (right). Parts of this figures have been adapted from [144,183,184,185].

## 8. Transmission between Plants and Control Strategies

Several strategies are used by viroids for its dissemination, some of which are facilitated by modern agricultural practices (Figure 5). The most effective strategy is the vegetative propagation of viroids through bulbs, tubers, rhizomes, or grafts [186,187], followed by mechanical transmission, especially during manipulations that involve mechanical damage to the plants, such as pruning and harvesting, and allows direct transmission by plant-to-plant contact or the use of contaminated agricultural machinery [35,186,187]. The presence of viroids in harvest residues, either fresh or processed, also poses a potential source for infection [188,189,190]. Much lower efficiency has been described for seed transmission of several viroids [35,187,191,192], as well as for infected pollen [191,193]. Spatial analyses of infection spread suggest viroid transmission through roots, which has been proven under experimental conditions [194,195,196]. The spread of viroid-like RNAs through parasitic plants and phytopathogenic fungi has also been proposed [179,181,197,198]. Insects are potential vectors of transmission, possibly by spreading infected pollen [192], and certain insect species could mediate the direct transmission of viroids of both families between plants [198,199,200], although the efficiency of these transmissions seems to be very low and unimportant from an epidemiological point of view. It has been described that the efficiency improves with the transencapsidation of the viroid RNA with an insect-transmitted plant virus, probably given the adaptation of the virus to its vector and its ability to efficiently infect plant cells [201]. Natural animal practices, such as goats rubbing their horns against the bark of infected trees, may also contribute to long-range viroid spread between cultivated and wild plants [202].

Effective commercial methods for the control of viroid infections are currently lacking, relying only on good agronomic practices to prevent, detect and eradicate the infection. Additionally, several strategies have been proposed for the control of viroids, such as genetic improvement of resistant varieties [203,204], cross-protection with latent viroids [205,206], or the generation of resistant transgenic plants, including RNAi-based strategies [129,132,207].

## 9. Biotechnological Aspects of Viroids

General research on viroids has led to important discoveries in RNA and plant biology, as recently reviewed [208]. In addition, viroids can be useful biotechnological tools, as with plant viruses. The ELVd has been used to overproduce recombinant RNAs in *E. coli,* such as aptamers and long dsRNAs with insecticidal activity [102,103,209,210,211]. The insertion of the RNA of interest in a particular position of the ELVd (+) RNA still allows the hammerhead ribozymes self-processing and ligation by a tRNA ligase, which is co-expressed, generating chimeras in which the circular viroid scaffold, compact and possibly associated with the ligase, is responsible for increasing the half-life of the RNA of interest and its accumulation in the bacteria. Also, recombinant clones of this same viroid carrying plant-specific sequences have been recently shown useful for dissecting gene functions in eggplant [212]. The range of applications of the viroids of the family *Avsunviroidae* can be further expanded as they are the only known pathogens capable of efficiently entering the chloroplast, an organelle of biotechnological interest. For example, chloroplasts lack RNAi machinery; thus, dsRNAs accumulating in there are not processed by the plant, and their uptake by insects is not hindered. Viroids inducing dwarfing have also been proposed as molecular tools to improve the high-density planting of citrus trees [168].

## 10. Conclusions

Viroids are the smallest infectious agents known to date. Despite extreme simplicity in terms of size and lack of protein-coding-capacity, viroid RNAs complete a complex infectious cycle in the infected plants, which includes genome replication, subcellular, cell-to-cell and long-distance movement and counteraction of host defense, frequently inducing a disease. Since viroids’ discovery about 50 years ago, a lot of knowledge has been gathered to understand viroid biology, but a lot is still missing, and some intriguing questions are currently being faced by viroid researchers. What is viroids’ evolutionary origin? Are viroids widespread in other taxonomic groups outside higher plants? Are viroids definitively non-coding RNAs, or may they still encode some functional peptides? How do these naked RNAs survive in the hostile environment of an infected cell? Do viroid RNAs contain other unnoticed ribozyme activities? Is viroid intracellular trafficking more complex than expected, combining phases in different organelles? Can viroid mighty molecular features be further exploited for biotechnological applications? We trust these and some other intriguing questions about viroid biology will be answered by the current and next generations of viroid researchers.

## Figures and Tables

**Figure 1 biology-12-00172-f001:**
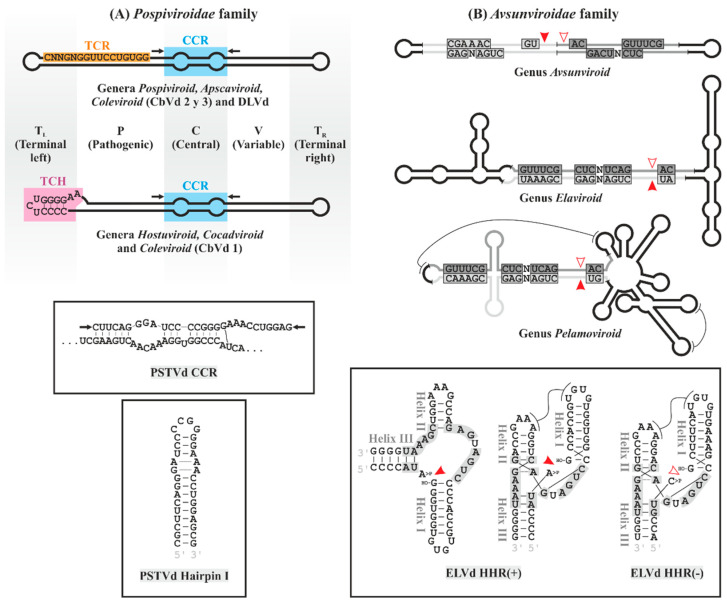
Structural characteristics of the viroids in the families *Pospiviroidae* and *Avsunviroidae*. (**A**) Members of the family *Pospiviroidae* adopt a rod-shaped secondary structure that has been functionally separated into five domains (TL, P, C, V and TR; differentially shaded). They contain conserved motifs: the features of the CCR (blue box) and the presence of TCR or TCH (orange and pink boxes, respectively) define the characteristics of each genus, as indicated. Together with the conserved sequence of the upper strand of the CCR, the flanking variable nucleotides (indicated by arrows) form an imperfect hairpin (hairpin I). Both the characteristic CCR sequence of PSTVd and the hairpin that forms are shown in the upper and lower inserts, respectively. (**B**) *Avsunviroidae* viroids adopt rod-shaped, branched or semibranched secondary structures (genus *Avsunviroid, Pelamoviroid* and *Elaviroid*, respectively). They contain conserved sequences of hammerhead ribozymes (HHR) that are functional in positive and negative strands (light and dark gray boxes, respectively, with the self-cleavage sites indicated by solid or empty arrowheads, respectively). In PLMVd, ‘kissing-loops’ tertiary interactions are indicated by lines. The insert includes the sequence of the HHR of ELVd with the classic representation that gives name to these ribozymes (left) next to the same HHR in both polarities according to the data of X-ray crystallography and NMR. Tertiary interactions between loops 1 and 2 are shown with lines. HO- and >P, 5′-hydroxyl and 2′,3′-phosphodiester groups, respectively; CCR, central conserved region; HHR, hammerhead ribozyme; N, any nucleotide; TCR, terminal conserved region; and TCH, terminal conserved hairpin.

**Figure 2 biology-12-00172-f002:**
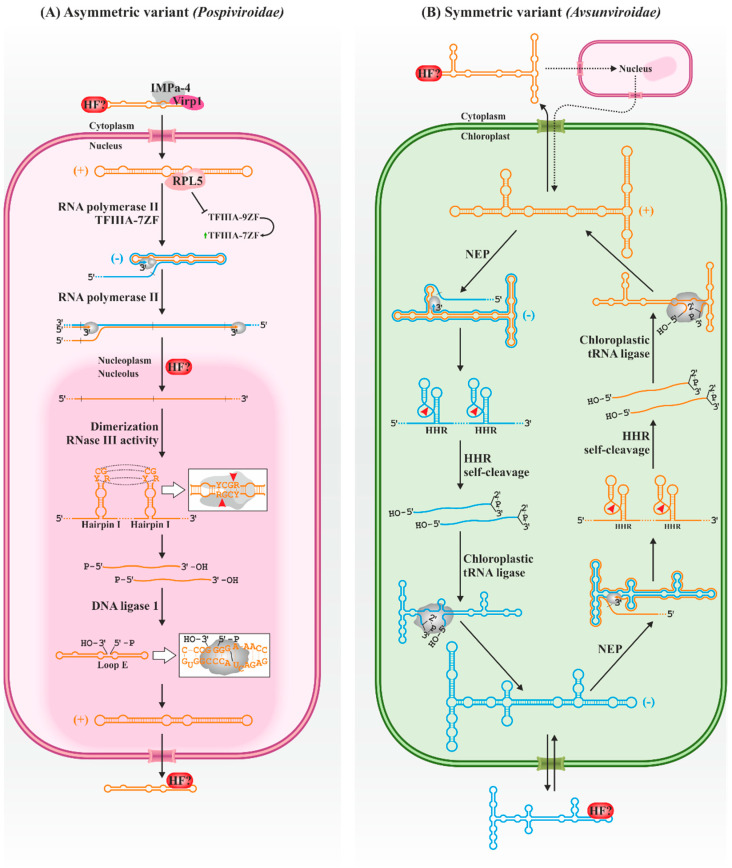
The rolling-circle mechanism in its (**A**) asymmetric and (**B**) symmetric variants is proposed for the replication of viroids of the families *Pospiviroidae* in the nucleus and *Avsunviroidae* in chloroplasts, respectively. In both cases, the positive and negative viroid RNA polarities are represented in orange and blue, respectively. Host proteins and viroid RNA motifs involved in the replicative cycle are indicated. Arrowheads indicate RNA cleavage sites. -P, -OH and >P, 5′-phosphate, 5′-hydroxyl and 2′,3′-phosphodiester groups, respectively; HF?, unknown host factor; HHR, hammerhead ribozyme; IMPa-4, importin alpha-4; NEP, nuclear-encoded chloroplastic DNA-dependent RNA polymerase; RPL5, ribosomal protein L5; TFIIIA-7ZF/-9ZF, transcription factor IIIA splicing variants with seven or nine zinc fingers, respectively; and Virp-1, bromodomain-containing protein 1.

**Figure 3 biology-12-00172-f003:**
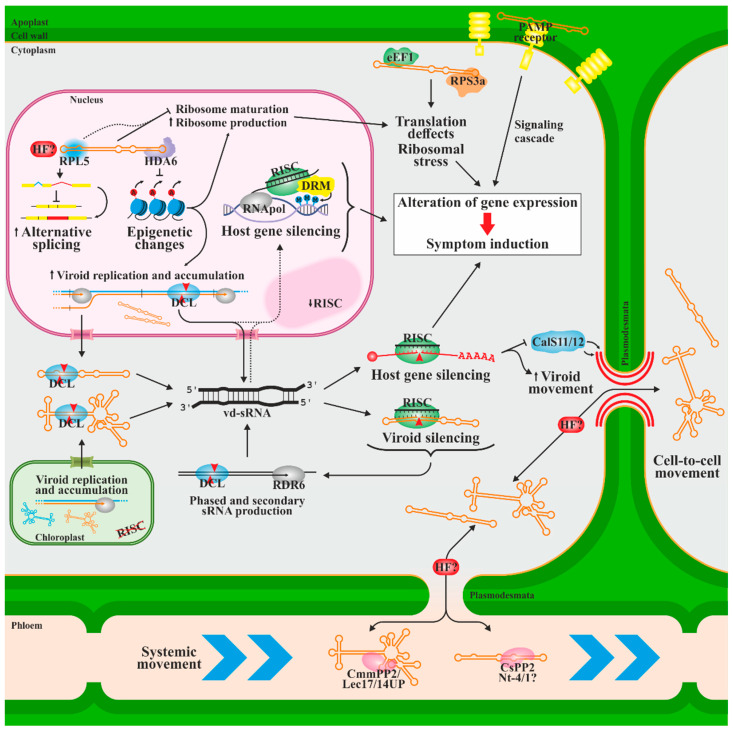
Proposed mechanisms of host defense responses, viroid pathogenesis and intercellular movement. Plant RNAi response is responsible for much of the viroid symptoms. dsRNA replicative intermediates and the cytoplasmic passage of viroids triggers the production of vd-sRNAs in plant cells. vd-sRNA-loaded RISC targets viroids and inhibits the expression of host genes containing complementary sequences post-transcriptionally by mRNA degradation and translation inhibition and possibly transcriptionally via RNA-directed DNA methylation. RDRs may transform sRNA fragments into additional DCL and RISC substrates. Viroid may also be recognized by cell membrane PAMP receptors stimulating plant innate immunity, resulting in the alteration of host gene expression. Additional interactions with proteins and host factors are responsible for global epigenetic changes, alternative splicing and interference with translational machinery, thus, are also involved in the development of symptoms. Viroids use plasmodesmata for proximal movement and phloem for systemic transport, likely interacting with specific (and in some cases unknown) host factors. RNAi response genes can increase intercellular movement. CalS11/CalS12, callose synthase 11 and 12, respectively; CmmPP2/Lec17/14UP, *Cucumis melo* phloem protein 2, phloem lectin 17 and uncharacterized protein of 14 kDa, respectively; DCL, Dicer-like protein; DRM, domains rearranged methylase; eEF1, eukaryotic elongation factor 1; HDA6, histone deacetylase; HF?, unknown host factor; Nt-4/1, *Nicotiana tabacum* 4/1 protein; PAMP, pathogen-associated molecular pattern; RISC, RNA-induced silencing complex; RDR6, RNA-dependent RNA polymerase 6; RNApol, RNA polymerase; RPL5, ribosomal protein L5; RPS3a, ribosomal protein S3a; and vd-sRNA, viroid-derived small RNAs.

**Figure 5 biology-12-00172-f005:**
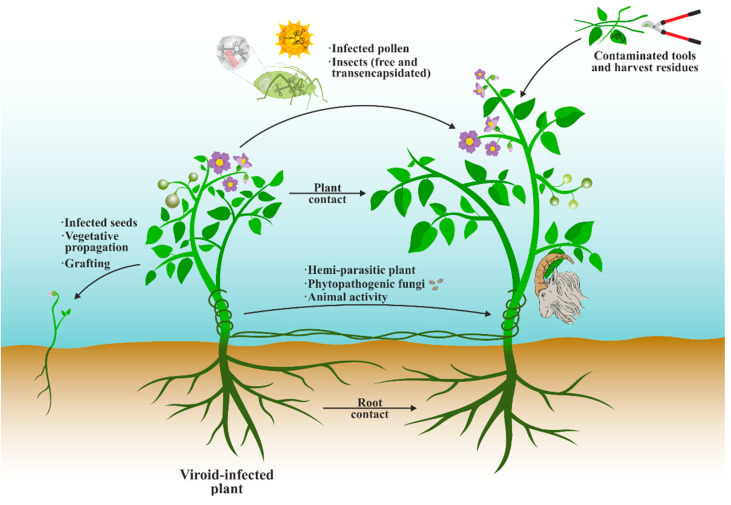
Some mechanisms of viroid transmission between plants.

**Table 1 biology-12-00172-t001:** ICTV taxonomic classification of viroids (2020). The 33 viroids are grouped into two families: *Pospiviroidae*, with five genera, and *Avsunviroidae*, with three genera. The type species of each genus is highlighted on a gray background. For each species, the abbreviation of its name is indicated.

Family Pospiviroidae		
Genus *Pospiviroid*	PSTVd	Potato spindle tuber viroid
	CEVd	Citrus exocortis viroid
	CSVd	Chrysanthemum stunt viroid
	CLVd	Columnea latent viroid
	IrVd-1	Iresine viroid 1
	PCFVd	Pepper chat fruit viroid
	TASVd	Tomato apical stunt viroid
	TCDVd	Tomato chlorotic dwarf viroid
	TPMVd	Tomato planta macho viroid
Genus *Hostuviroid*	HSVd ^1^	Hop stunt viroid
	DLVd	Dahlia latent virus
Genus *Apscaviroid*	ASSVd	Apple scar skin viroid
	ADFVd	Apple dimple fruit viroid
	AGVd	Australian grapevine viroid
	CBLVd ^1^	Citrus bent leaf viroid
	CDVd ^1^	Citrus dwarfing viroid
	CVd-V ^1^	Citrus viroid V
	CVd-VI ^1^	Citrus viroid VI
	GYSVd-1	Grapevine yellow speckle viroid 1
	GYSVd-2	Grapevine yellow speckle viroid 2
	PBCVd	Pear blister canker viroid
Genus *Cocadviroid*	CCCVd	Coconut cadang-cadang viroid
	CTiVd	Coconut tinangaja viroid
	CBCVd ^1^	Citrus bark cracking viroid
	HLVd	Hop latent viroid
Genus *Coleviroid*	CbVd-1	Coleus blumei viroid 1
	CbVd-2	Coleus blumei viroid 2
	CbVd-3	Coleus blumei viroid 3
**Family *Avsunviroidae***		
Genus *Avsunviroid*	ASBVd	Avocado sunblotch viroid
Genus *Pelamoviroid*	PLMVd	Peach latent mosaic viroid
	CChMVd	Chrysanthemum chlorotic mottle viroid
	AHVd	Apple hammerhead viroid
Genus *Elaviroid*	ELVd	Eggplant latent viroid

^1^ Names of some viroid species have been re-established by ICTV. This particularly affects citrus viroids, such as CBCVd (formerly citrus viroid IV), CBLVd (formerly citrus viroid I), CDVd (formerly citrus viroid III), HSVd (formerly citrus viroid II) and CVd-V and VI (formerly citrus viroid-OS).

## Data Availability

Not applicable.

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
