# Peer review of "Viroids: Non-Coding Circular RNAs Able to Autonomously Replicate and Infect Higher Plants"

_biology, 2023, doi:10.3390/biology12020172_

Round 1
Reviewer 1 Report
Generally, I would really like to see this published asap. This is a good source for the latest info on viroid research discussed in the context of earlier findings. It is citing a lot of recent findings and I do not miss anything important. It is well balanced across the whole dimension of viroid biology.
If possible, I would like to see a way to highlight the research gaps. They are well described but maybe they can be set to italic or briefly restated in the conclusion – maybe in bullet points? This is only a suggestion for the authors AND editor.
Dear authors, thank you for your work!
Line 20 … thereby inducing diseases in some host plants. (simpler, better understandable)
Line 21 omit “and early 70s” since this is the starting point also mentioned in the introduction
Line 30-31 … ,also for the initiation of potential biotechnological applications of viroid molecules. (simpler)
Line 60 Suggestion: It think the origin of viroids can be structured into two overall directions – viroids evolved (out of or through interaction with their host plant or plant viruses) OR viroids are ancient entities (precursor of life itself). I suggest that such a clear differentiation makes the paragraph clearer. But it’s just a suggestion
Line 63 “amazing” > “great” or “larger than expected” or else to make it more specific
Figure 1 Suggestion: I would appreciate if the former names would be included in this figure/table – The names are not consistently used in the literature e.g. CDVd was known as Citrus Viroid III or HSVd was also called Citrus Viroid II etc – I would find this helpful and it can be easily implemented while also reflecting some viroid research history …
Figure 1 is a table, please change
Line 153 How does NF? Abbreviate unknown host factor? rather HF?
Line 155 I would like all abbreviations to be shown here to make the figure clearer and easier to read
Figure 2 B: The path for the asunviroid through the nucleus is based on a hypothesis based on evidence provided by 2 papers 74, 75 – correct? I would suggest indicating that with a “?” on the pathway arrows and offer an alternative pathway directly into the chloroplast. I guess a labelling experiment with extracted chloroplasts could support this hypothesis. I am not aware of such an study, but also I did not look into it yet. I just worry that this figure would be taken as solid, while this aspect seems to be still rather open in contrast to the other hypothesis integrated in this figure. I suggest highlighting this as one of the open questions you mentioned in the conclusion (Great figure by the way).
Line 201 – to what literature are you referring when writing “in the current accepted model …? Please add
Line 241 … the involvement of the structural motifs cannot be ruled out
Figure 4. Again there are some unexplained abbreviations and the NF>HF? Issue. Also instead of Host TGS? Would Post-transcriptional gene silencing or simply gene silencing be sufficient for the interaction shown? Also the elongation factor eEF1 or RPS3a are not explained in the figure and only are mentioned several pages later. It took me a while to find the connection between text and Figure 4, so please explain briefly in the figure and indicate the figure number in the text.
Line 325 … remains a mystery … better remains unsolved or unanswered. I think this can or will be addressed by experiments in the near future if only one finds the time and money
Line 329 … Regarding this chapter - maybe it would be more balanced if here the term TsnRNA and the use of viroids as dwarfing agents would also be mentioned and discussed here, since certainly some viroid-associated plant responses can be beneficial https://pubmed.ncbi.nlm.nih.gov/35744662/ I guess it would fit either at the beginning of the chapter or/and in line 415ff when talking about induction of the immune response following pathogeneic viroids. Lavene et al has shown that the immune response is lacking for non-pathogenic viroids. Also it would fit line 429 …
Line 345 please properly introduce the abbreviation of TGS if necessary, at all. I think RNAi is a sufficient term for this article also for Figure 4, why differentiate between TGS and PTGS, since the latter abbreviation is not even mentioned again …
Line 351 abbreviation AGO not introduced, maybe just argonaut?
Line 370 PC = peach calico is an unnecessary abbreviation since only used twice after introduction
Line 386 TSG (as well as TGS) is only introduced but not used at all, thus can be omitted
Line 399 sometimes the full name and sometimes only the family name of an cited author is written in the text. Please be consistent and in line with the journals guidelines
Line 422 what about the viroid-like RNA named Dendrobium viroid (DVd)? Exclude because it is not yet accepted as viroid by the taxonomy group? https://doi.org/10.1016/j.virusres.2021.198626
Line 423 CBVd4-6 are not yet recognised as viroids by the taxonomy group, maybe as a work around write Coleus blumei viroids
Line 440 CVd-IV has been called CBCVd in Figure 1 – please be consistent
Line 455 please add for completing the risks “The presence of viroids in harvest residues, either fresh or processed, also poses a potential source for infection https://link.springer.com/article/10.1007/s10658-021-02344-2 and maybe also https://bsppjournals.onlinelibrary.wiley.com/doi/full/10.1111/ppa.12729
Line 463 consider adding goats as example for unexpected viroid sources ? https://link.springer.com/article/10.1007/BF03029972
Line 456 there is a recent study of fungi on apple also showing ASSVd infections https://www.mdpi.com/2073-4409/11/22/3686
Figure 6 – here an aphid would fit better then a fly. I am not aware of flies playing a role in any viroid pathogen interaction thus this may be misleading. Aphids, however, are the vectors in the literature you cited
Reviewer 2 Report
Viroids are very important pathogens in agriculture and therefore a revision of the knowledge that we have about them is always necessary.The review is clear and concise and I am sure it will help senior researchers as well as thesis or degree students. The molecular description and the mechanisms of the biology of viroids are admirable, but I think that sections 6, 7 and 8 could be elaborated a little more. I think it is necessary to go a little deeper into the ecology and epidemiology of viroids.
